# Neural MMO: A massively multiplayer game environment for intelligent agents

## Abstract

We present an artificial intelligence research platform inspired by the human game genre of MMORPGs (Massively Multiplayer Online Role-Playing Games, a.k.a. MMOs). We demonstrate how this platform can be used to study behavior and learning in large populations of neural agents. Unlike currently popular game environments, our platform supports persistent environments, with variable number of agents, and open-ended task descriptions. The emergence of complex life on Earth is often attributed to the arms race that ensued from a huge number of organisms all competing for finite resources. Our platform aims to simulate this setting in microcosm: we conduct a series of experiments to test how large-scale multiagent competition can incentivize the development of skillful behavior. We find that population size magnifies the complexity of the behaviors that emerge and results in agents that out-compete agents trained in smaller populations.

## 1 Introduction

Life on Earth can be viewed as a massive multiagent competition. The cheetah evolves an aerodynamic profile in order to catch the gazelle, the gazelle develops springy legs to run even faster: species have evolved ever new capabilities in order to outcompete their adversaries.

The success of biological evolution has inspired many attempts to emulate it in silico, ranging from genetic algorithms that bear only loose resemblance to natural processes, to full-blown simulations of "artificial life". A recurring question has been: at what level of abstraction should we simulate the competitive game of life?

In recent years, the field of deep reinforcement learning (RL) has embraced a related approach: train algorithms by having them compete in simulated games (Silver et al., 2016; OpenAI; Jaderberg et al., 2018). Such games are immediately interpretable and provide easy metrics derived from the game's "score" and win conditions. However, popular game benchmarks are currently still limited: they typically define a narrow, episodic task, with a small fixed number of players. In contrast, life on Earth involves a persistent environment, an unbounded number of players, and a seeming "open-endedness", where ever new and more complex species emerge over time, with no end in sight (Stanley et al., 2017).

Our aim is to develop a simulation platform (see Figure 5) that captures important properties of life on Earth, while also borrowing from the interpretability and abstractions of human-designed games. To this end, we turn to the game genre of Massively Multiplayer Online Role-Playing Games (MMORPGs, or MMOs for short). These games involve a large, variable number of players competing to survive and prosper in persistent and far-flung environments. Our platform simulates a "Neural MMO" – an MMO in which each agent is a neural net that learns to survive using RL.

We demonstrate the capabilities of this platform through a series of experiments that investigate emergent complexity as a function of the number of agents and species that compete in the simulation. We find that large populations act as competitive pressure that encourages exploration of the environment and the development of skillful behavior. In addition, we find that when agents are organized into species (share policy parameters), each species naturally diverges from the others to occupy its own behavioral niche. Upon publication, we will opensource the platform in full.

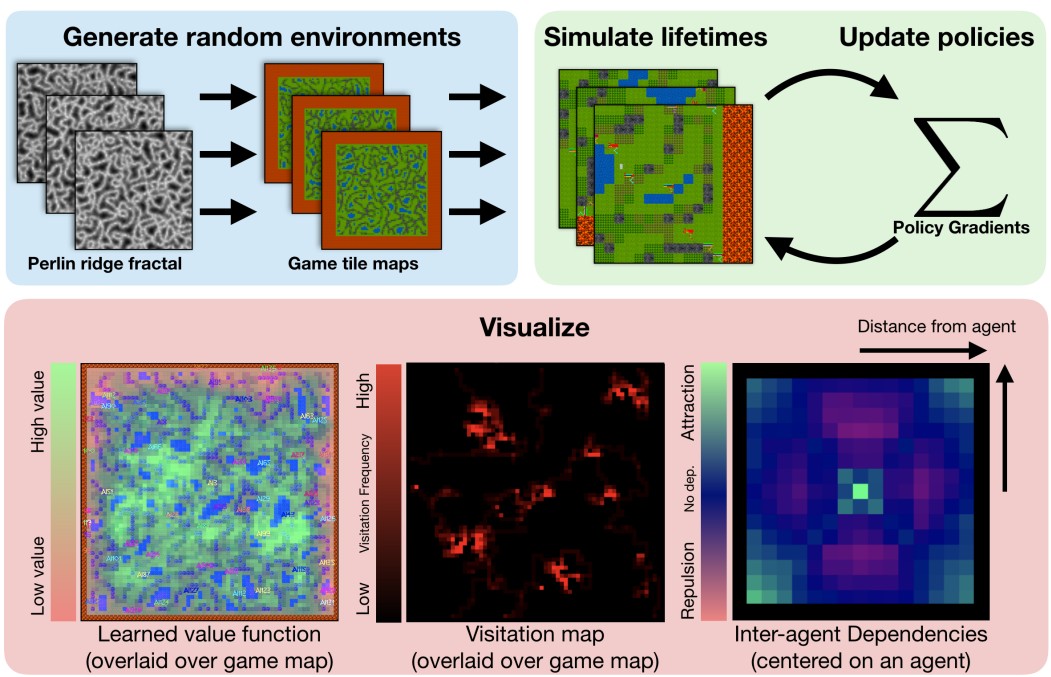

Figure 1: Neural MMO pipeline. We alternate between collecting experience across 100 procedurally generated worlds and updating agents' parameters via policy gradients. Test time visualization provides insight into the learned policies through value function estimates, map tile visitation distribution, and agent-agent dependencies.

## 2 BACKGROUND AND RELATED WORK

**Multiagent Reinforcement Learning**   Multiagent reinforcement learning has received increased attention in recent years (Lowe et al., 2017; Mordatch & Abbeel, 2017; Bansal et al., 2017; Lanctot et al., 2017; Yang et al., 2018a; Zheng et al., 2017; OpenAI; Jaderberg et al., 2018). Unlike single agent environments often well modeled by classic Markov Decision Processes (MDPs), multiagent interaction often introduces nonstationarity in the transition dynamics of the environment due to the continual learning and co-adaptation of other agents.

While previous work has attempted to analyze emergent complexity in interactions of groups of 2-10 agents (Bansal et al., 2017; OpenAI; Jaderberg et al., 2018), we focus on large populations of agents in a complex world. Zheng et al. (2017) also analyze behavior of many agents, but the task setting is considerably simpler, and agents are directly rewarded for foraging or fighting. In contrast, our work focuses on complexity in a diverse environment where agents are only rewarded for survival; doing so requires maintaining their health through food and water while navigating partially obstructed terrain and with various forms of combat. Yang et al. (2018b) also simulate populations scaling to millions of learning agents, but they focus on predator-prey population dynamics in this setting.

**Artificial Life**   "Artificial life" projects aim to simulate the complexity of biological life (Langton, 1997), often framed as a multiagent competition. This setting can lead to the development of skilled behaviors (Yaeger, 1994) and capable morphologies (Sims, 1994). We consider the setting of coevolution (Ficici & Pollack, 1998) in which agents coadapt alongside others.

Our platform can simulate a kind of artificial life, but at a relatively high level of abstraction. While some basic features of our environment (movement, food, water) are similar to those in (Hernández-Orallo et al., 2011; Strannegrd et al., 2018), our environment is grounded in the established game genre of MMOs. Unlike most past work in this area, our platform is built around deep reinforcement learning, where each agent is a neural net trained with distributed policy gradients, and includes tools for visualizing properties specific to this setting.

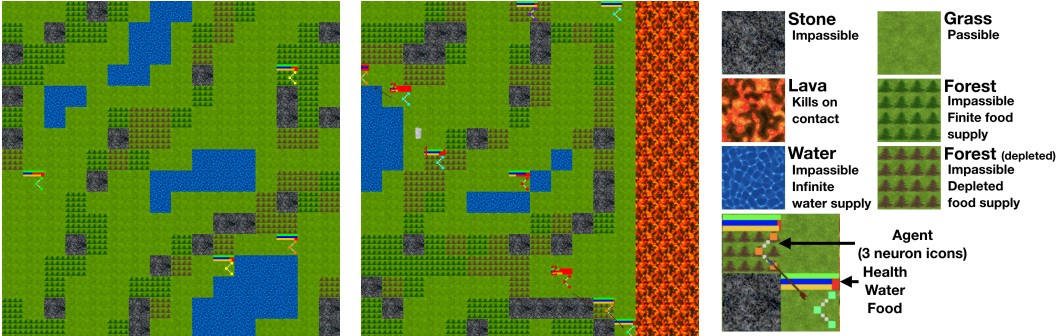

Figure 2: Left: Foraging agents learn to efficiently balance their food and water levels while competing with other agents for resources. Center: Combat agents learn the latter while also balancing melee, range, and mage attack styles to engage with and outmaneuver other agents. Right: Graphics key for tiles and agents.

**Game Platforms for Intelligent Agents**  The Arcade Learning Environment (ALE) (Bellemare et al., 2013) and Gym Retro (Nichol et al., 2018) provide 1000+ limited scope arcade games most often used to test individual research ideas or generality across many games. Strictly better performance at a large random subset of games is a reasonable metric of quality, and strikingly divergent performance can motivate further research into a particular mode of learning. For example, Montezumas Revenge in ALE is difficult even with reasonable exploration because of the associative memory required to collect and use keys. However, recent results have brought into question the overall complexity each individual environment  (Cuccu et al., 2018) and strong performance in such tasks (at least in those not requiring precise reflexes) is not particularly difficult for humans.

More recent work has demonstrated success on multiplayer games including Go (Silver et al., 2016), the Multiplayer Online Battle Arena (MOBA) DOTA2 (OpenAI), and Quake 3 Capture the Flag (Jaderberg et al., 2018). Each of these projects has advanced our understanding of a class of algorithms. However, these games were limited to 2-12 players, are round based on the order of an hour, lack persistence, and lack the game mechanics supporting large persistent populations – there is still a large gap in environment complexity compared to the real world.

**MMORPGs**  The game genre we focus on in this paper is MMORPGs, which are role-playing games (RPGs) in which many human players take part. RPGs, such as Pokemon and Final Fantasy, involve reasoning over up to hundreds of hours of persistent gameplay – a much longer time horizon than in MOBAs. Like the real world, RPGs confront the player with problems that have many valid solutions, and choices that have long term consequences.

MMOs are the (massively) multiplayer analogs to RPGs.  They are typically run across several **servers**, each of which contains a copy of the **environment** and supports hundreds to millions of concurrent **players**. Good MMOs involve a curriculum of challenges that require increasingly clever usage of the game systems. Early game content is accessible to new players, but skills required for late game content are inaccessible (and often incomprehensible) to those not intimately familiar with the game. Players have to acquire resources and "level up" in order to reach more advanced stages of the game. **Such a curriculum is present in many game genres, but only MMOs contextualize it within social and economic structures approaching the scale of the real world.**

## 3  NEURAL MMO

We present a persistent and massively multiagent environment that defines foraging and combat systems over procedurally generated maps (see Appendix). **The purpose of the platform is to discover *game mechanics* that support complex behavior and *agent populations* that can learn to make use of them.** We follow the iterative development cycle of human MMOs: **developers create balanced mechanics while players maximize their skill in utilizing them**. The initial configurations of our systems are the results of several iterations of balancing, but are by no means fixed: every numeric parameter presented is editable within a simple configuration file.

| Parameter | Value | Notes |
|---|---|---|
| Training Algorithm | Policy GradientsWilliams (1992) | + Value function baseline |
| Adam Parameters | lr=1e-3 | Pytorch Defaults |
| Weight Decay | 1e-5 | Training stability is sensitive to this |
| Entropy Bonus | 1e-2 | To stabilize training; possibly redundant |
| Discount Factor | 0.99 | No additional trajectory postprocessing |

A map consists of a set of discrete tiles, which are positions an agent can occupy. On each step of the simulations, agents may move one tile North/South/East/West. Agents competes for food tiles while periodically refilling their water supply from infinite water tiles. They may attack each other using any of three attack options, each with different damage values and trade offs.

The environment assumes only that agents receive local game state and output a decision. The environment is agnostic to the source of that decision, be it a neural network or a hardcoded algorithm. We have tested up to 100M agent trajectories (lifetimes) on 100 cores in 1 week. The code base already contains additional module code for trade, gathering, crafting, item systems, equipment, communication, and trade to name a few. We are actively balancing and integrating these into the neural API.

## 4 ARCHITECTURE AND TRAINING

**Input** Agents observe local game state–all tiles within a fixed $L_1$ distance of their current position, including tile terrain types and the visible properties of occupying agents. This is an efficient equivalent representation of what a human sees on the screen without requiring rendering.

**Output** Agents output action choices for the next time step ("game tick"). For the experiments below, actions consist of one movement and one attack. As described in Framework, movement options are: North, South, East, West, Pass (no movement). Attack options are: Melee, Range, Mage. This is purely flavor; for those not familiar with MMOs, each attack option simply applies a preset amount of damage at a preset effective distance. The environment will attempt to execute both actions. Invalid actions, such as moving into a stone wall, are ignored.

Our policy architecture is detailed in the Appendix. We provide a simple preprocessor that embeds and flattens this stimulus into a single fixed length environment vector and a list of entity embeddings. We apply a linear layer to the preprocessed embeddings followed by three output heads for movement, attacks. There is also a standard value head which is trained to predict the discounted expected lifetime of the agent. Each head is also a linear layer. New types of action choices can be included by adding additional heads. We train with simple policy gradients plus a value baseline. Agents receive only a stream of reward 1. Rewards are postprocessed by a discounting factor, producing returns equal to a discounted estimate of the agent's time until death. We found it possible to obtain good performance without discounting, but training was less stable.

## 5 EXPERIMENTS

We present an initial series of experiments using our platform to explore multiagent interactions in large populations. We find that agent competence scales with population size. In particular, increasing the maximum number of concurrent players ($N_{ent}$) magnifies exploration and increasing the maximum number of populations with unshared weights ($N_{pop}$) magnifies niche formation. Agents are sampled uniformly from a number of "populations"–identical architectures with unshared weights. This is for efficiency–see technical details.

Technical details We run each experiment using 100 worlds. We define a constant $C$ over the set of worlds $W$. For each world $w \in W$, we uniformly sample a $c \in (1, 2, ...C)$. We define "spawn cap" such that if world $w$ has a spawn cap $c$, the number of agents in $w$ cannot exceed $c$. In each world $w$, one agent is spawned per game tick provided that doing so would exceed the spawn cap $c$ of $w$. Ideally, we would fix $N_{ent} = N_{pop}$, as is the case in standard MMOs (humans are independent networks with unshared weights). However, this incurs sample complexity proportional to number of populations. We therefore share parameters across groups of up to 16 agents for efficiency.

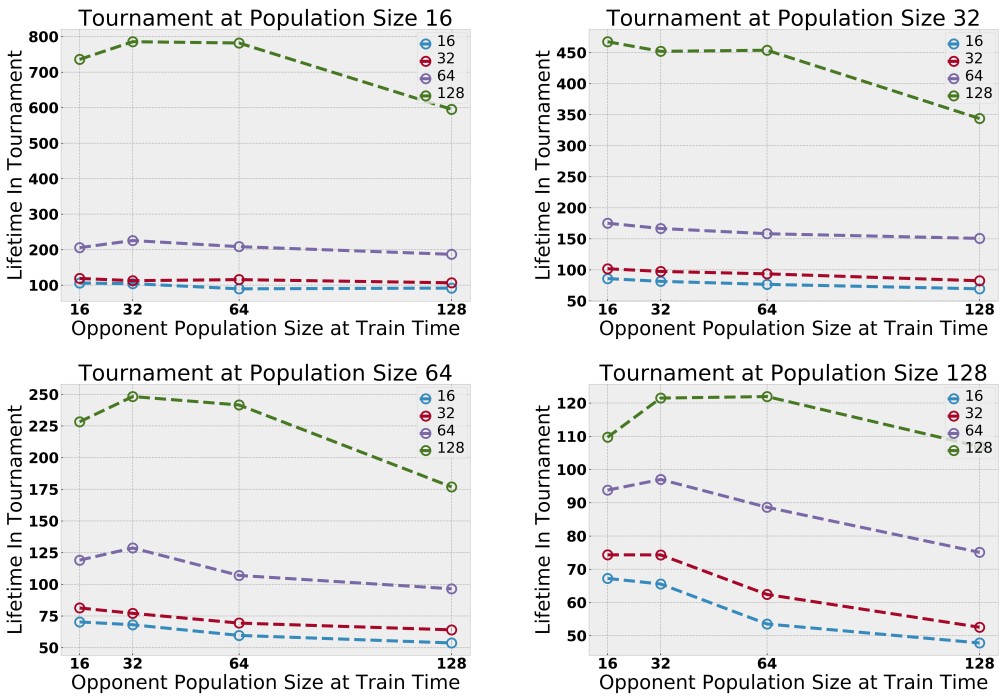

Figure 3: Maximum population size at train time varies in 16, 32, 64, 128. At test time, we merge the populations learned in pairs of experiments and evaluate lifetimes at a fixed population size. Regardless of population size at test time, we find that agents trained in larger populations outperform agents trained in small populations

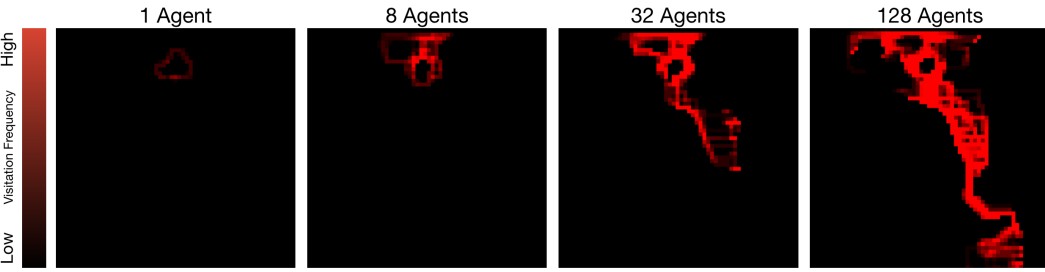

Figure 4: Population size magnifies exploration: agents spread out to avoid competition.

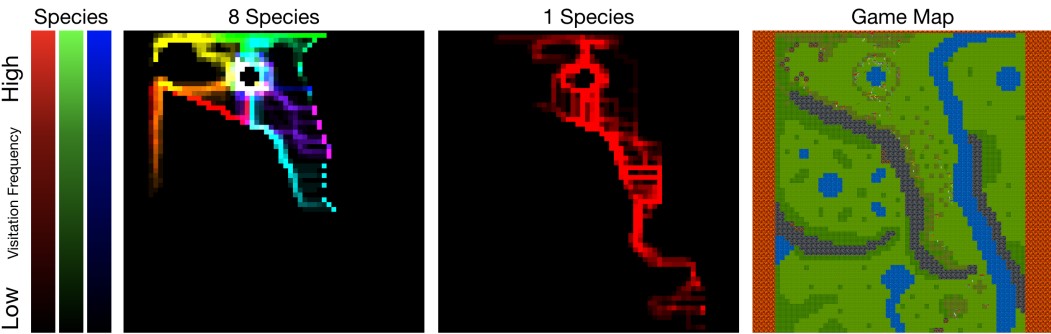

Figure 5: Populations count (number of species) magnifies niche formation. Visitation maps are overlaid over the game map; different colors correspond to different species. Training a single population tends to produce a single deep exploration path. Training 8 populations results in many shallower paths: populations spread out to avoid competition among species.

## 5.1 SERVER MERGE: ELO FACILITATES GAME MECHANICS AND INTERPRETABILITY

We perform four experiments to evaluate the effects on foraging performance of training with larger and more populations. For each experiment, we fix $N_{pop}$ and a spawn cap [1]. These are paired from (1, 2, 4, 8) and (16, 32, 64, 128). We train for a fixed number of trajectories per population 9.

Evaluating the influence of these variables is nontrivial. The task difficulty is highly dependent on the size and competence of populations in the environment: mean agent lifetime is not comparable among experiments. Furthermore, there is no analog procedure for evaluating relative player competence among MMO servers. However, MMO servers sometimes undergo merges whereby the player bases from multiple servers are placed within a single server. As such, we propose tournament style evaluation in order to directly compare policies learned in different experiment settings. Tournaments are formed by simply concatenating the player bases of each experiment. Results are shown in Figure 3: we vary the maximum number of agents at test time and find that agents trained in larger settings consistently outperform agents trained in smaller settings.

When we introduce the combat module as an additional learnable mode of variation on top of foraging, we observe more interesting policies as a result. Agent actions become strongly coupled with the states of other agents. As a sanity, we also confirm that all of the populations trained with combat handily outperform all of the populations trained with only foraging.

To better understand theses results, we decouple our analysis into two modes of variability: maximum number of concurrent players ($N_{ent}$) and maximum number of populations with unshared weights ($N_{pop}$). This allows us to examine the effects of each factor independently. In order to isolate the effects of environment randomization, which also encourages exploration, we perform these experiments on a fixed map. Isolating the effects of these variables produces more immediately obvious results, discussed below. We briefly examine the randomized setting in Discussion.

## 5.2 $N_{ent}$: MULTIAGENT COMPETITION MAGNIFIES EXPLORATION

In the natural world, competition between animals can incentivize them to spread out in order to avoid conflict. We observe that overall exploration scales with number of concurrent agents, with no other variable factors (see Figure 4; the map used is shown in Figure 5). Agents learn to explore only because the presence of other agents provides a natural and beneficial curriculum for doing so.

## 5.3 $N_{pop}$: MULTIAGENT COMPETITION MAGNIFIES NICHE FORMATION

We find that, given a sufficiently large and resource rich environment, different populations of agents tend to separate to avoid competing with other populations. Both MMOs and the real world often reward masters of a single craft more than jacks of all trades. From Figure 5, specialization to particular regions of the map scales with number of populations. This suggests that the presence of other populations force agents to discover a single advantageous skill or trick. That is, increasing the number of populations results in diversification to separable regions of the map. As entities cannot out-compete other agents of their population with shared weights, they tend to seek areas of the map that contain enough resources to sustain their population. Regions that are difficult to get to or otherwise unoccupied are especially desirable; this is revealed by observing value maps over time.

## 6 DISCUSSION

### 6.1 MULTIAGENT COMPETITION IS A CURRICULUM MAGNIFIER

Jungle climates produce more biodiversity than deserts. Deserts produce more biodiversity than the tallest mountain peaks. To current knowledge, Earth is the only planet to produce life at all: the initial conditions for formation of intelligent life are of paramount importance. The same holds true in simulation: human MMOs mirror this phenomenon. Some games produce more complex and engaging play than others–those most successful garner large and dedicated playerbases that come to understand the game systems better than the developers. Feedback helps drive development and expansion over a period of years, but the amount of effort required to create that initial seed is large.

---

[1]The maximum number of concurrent agents, see technical details

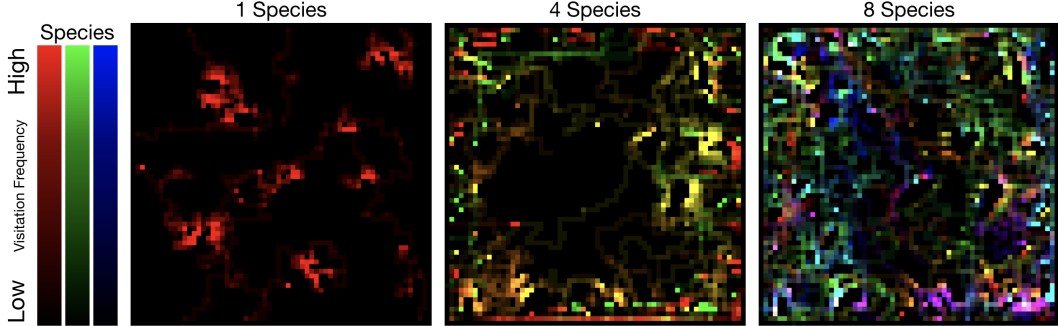

Figure 6: Exploration maps in the environment randomized settings. From left to right: population size 8, 32, 128. All populations explore well, but larger populations with more species develop more robust policies, utilize the map more efficiently, and thereby do better in tournaments.

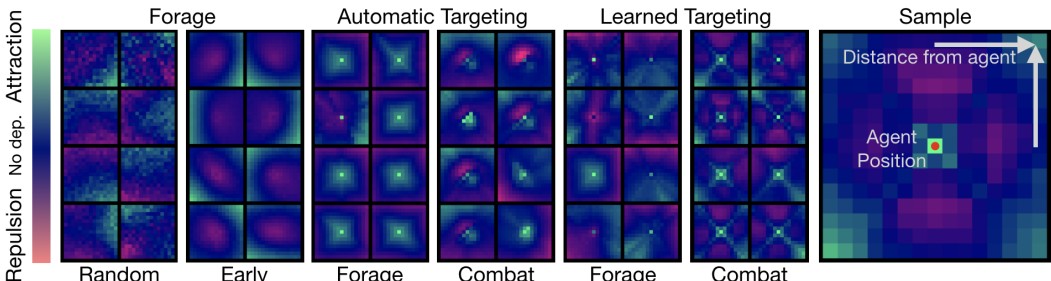

Figure 7: Agents learn to depend on other agents. Each square map shows the response of an agent of a particular species, located at the square's center, to the presence of agents at any tile around it. Random: dependence map of random policies. Early: "bulls eye" avoidance maps learned after only a few minutes of training. Additional maps correspond to foraging and combat policies learned with automatic targeting (as in tournament results) and learned targeting (experimental, discussed in Additional Insights). In the learned targeting setting, agents begin to fixate on the presence of other agents within combat range, as denoted by the central square patterns.

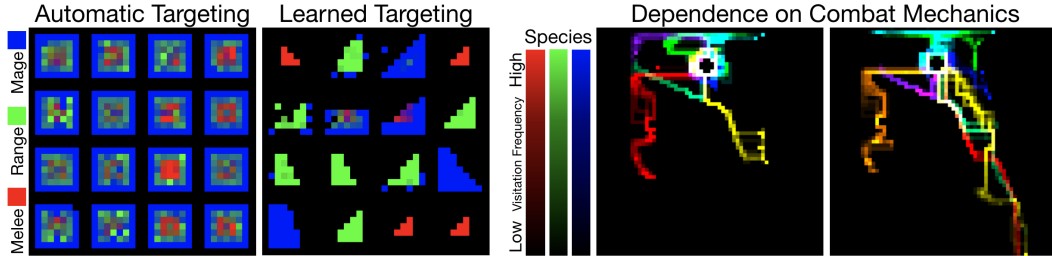

Figure 8: Attack maps and niche formation quirks. Left: combat maps from automatic and learned targeting. The left two columns in each figure are random. Agents with automatic targeting learn to make effective use of melee combat (denoted by higher red density). Right: noisy niche formation maps learned in different combat settings with mixed incentives to engage in combat.

It is unreasonable to expect pure multiagent competition to produce diverse and interesting behavior if the environment does not support it. This is because **multiagent competition is a curriculum *magnifier*, not a curriculum**. The multiagent setting is interesting because learning is responsive to competitive and collaborative pressures of other learning agents–but the environment must support and facillitate such pressures in order for multiagent interaction to drive complexity.

There is room for debate among reasonable individuals as to theoretical minimum complexity seed environment required to produce complexity on par with that of the real world. However, this is not our objective, and we do not have a reasonable estimate of this quantity. We have chosen to model our environment after MMOs, even though they may be more complicated than the minimum required environment class, because they are known to support the types of interactions we are interested in while maintaining engineering and implementation feasibility. This is not true of any other class environments we are aware of: exact physical simulations are computationally infeasible, and previously studied genres of human games lack crucial elements of complexity (see Background). While some may see our efforts as cherrypicking environment design, we believe this is precisely the objective: developers cherrypick game design decisions to support complexity commensurate with engaging play at the level of general human intelligence. The player base uses these design decisions to create strategies far beyond the imagination of the developers.

## 6.2 ENVIRONMENT RANDOMIZED EXPLORATION

The trend of increasing exploration with increasing entity number is clear when training on a single map as seen in Figure 4, 5, but it is more subtle with environment randomization. From Figure 6, all population sizes explore adequately. We believe that this is because "exploration" as defined by map coverage is not as difficult a problem as developing robust policies. As demonstrated by the Tournament experiments, smaller populations learn brittle policies that do not generalize to scenarios with more competitive pressure–even against a similar number of agents.

## 6.3 AGENT-AGENT DEPENDENCIES

We visualize agent-agent dependencies in Figure 7. We fix an agent at the center of a hypothetical map crop. For each position visible to that agent, we fake another agent and compute the value function estimate of the resultant pair. We find that agents learn policies dependent on those of other agents in both the foraging and combat environments.

## 6.4 ADDITIONAL INSIGHTS

We briefly detail several miscellaneous investigations and subtle points of interest in Figure 8. First, we visualize learned attack patterns of agents. Each time an agent attacks, we splat the attack type to the screen. There are a few valid strategies as per the environment. Melee is intentionally overpowered: learning to utilize it at close range serves as a sanity check. This also cautions agents to keep their distance, as the first to strike wins. From Figure 8 and observation of the policies, we find that this behavior is learned when targeting is automated.

Our final set of experiments prescribes targeting to the agent with lowest health. Jointly learning attack style selection and targeting requires an attentional mechanism to handle variable number of visible targets. We have experimented with this, but results are not yet numerically stable. Solving this is likely to increase policy complexity, but policies learned using automatic targeting are still compelling: agents learn to strafe at the edge of their attack radius, attacking opportunistically.

Second, a note on tournaments. We equate number of trajectories trained upon as a fairest possible metric of training progress. We experimented with normalizing batch size but found that larger batch size always leads to more stable performance. Batch size is split among species: larger populations outperform smaller populations even though the latter are easier to train.

Finally, a quick note on niche formation. Obtaining clean visuals is dependent on having an environment where interaction with other agents is unfavorable. When this is not the case, niche formation may still occur in another space (e.g., diverse attack policies). This is the importance of niche formation–we expect it to stack well with population-based training (Jaderberg et al., 2017) and other such methods that require sample diversity.

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

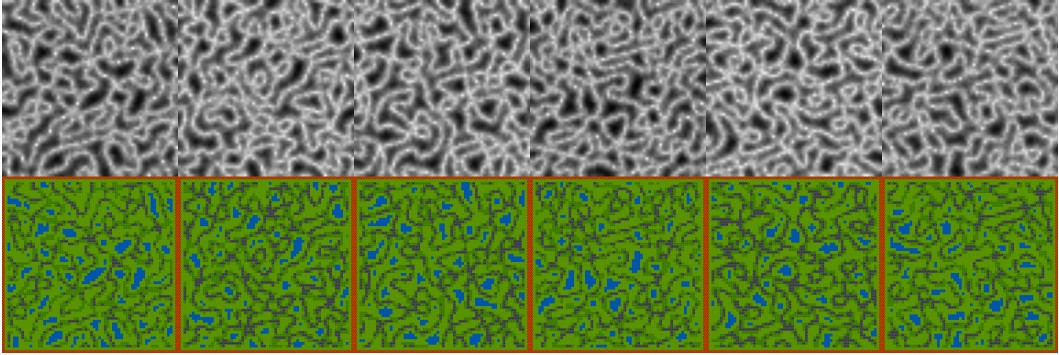

Figure 9: We procedurally generate maps by thresholding an 8 octave Perlin (Perlin, 1985) ridge fractal. We map tile type to a fixed range of values

## APPENDIX

## 7 FRAMEWORK

**This section contains full environment details that are useful but not essential to understanding the base experiments.** All parameters in the following subsystems are configurable; we provide only sane defaults obtained via balancing.

### 7.1 ENVIRONMENT TILES

A grid of tiles[2] represents game state. Agents may move one tile North/South/East/West each game tick. They can also choose to Pass (i.e. not make a movement action). Each tile object contains:

- **Terrain type** $\in \{grass, forest, stone, water, lava\}$. Lava tiles kill agents upon contact. Stone and water tiles are impassable.
- **Occupying agents:** a reference mapping of agents standing on the tile.

### 7.2 FORAGING

The foraging system implements gathering based survival by introducing:

- **Food:** initialized to 32, decremented 1 per tick, incremented 1 by occupying forest tiles, which contain 1 food each. Once the resources of a particular tile are consumed, they regenerate probablistically over time.
- **Water:** decremented once per tick, incremented by standing adjacent to a water tile.
- **Health:** decremented once per tick if either food or water is zero. Incremented if both food and water are above a threshold.

A snapshot is shown in Figure 2. These definitions of resources impose a carrying capacity. This incurs an arms race of exploration strategies in populations of agents above the carrying capacity: survival is trivial with a single agent, but requires intelligent exploration in the presence of competition attempting to do the same.

### 7.3 COMBAT

The combat system implements three different attack "styles":

- **Melee:** Inflicts 10 damage at 1 range

---

[2]Many MMOs use an internal tile grid representation and smooth animations during rendering. This enables servers to tick at a rate of once or twice per second without looking unnatural. We adopt this model in our work.

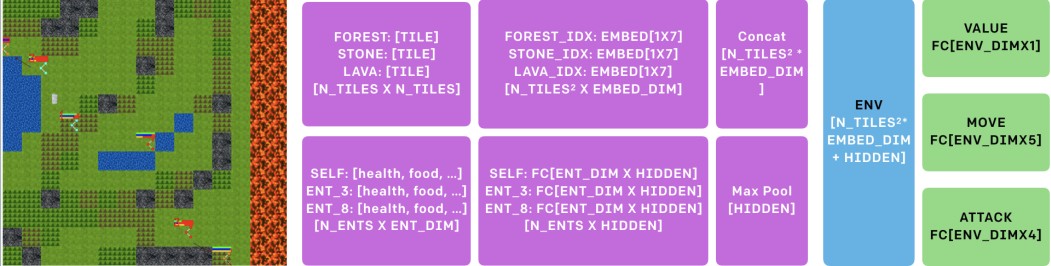

Figure 10: Agents observe their local environment. The model embeds this observations and computes actions via corresponding value, movement, and attack heads. These are all small fully connected networks with 50-100k parameters.

- **Ranged:** Inflicts 2 damage at 1-2 range
- **Mage:** Inflicts 1 damage at 3 range and freezes the target in place, preventing movement (but not attacks) for two ticks

A snapshot is shown in Figure 2. These definitions of combat impose trade offs in each style. Melee combat is high risk high return. Ranged combat produces less risky but more prolonged conflicts. Mage combat does little damage, but allows agents to retreat or cut off opponents' escape by freezing them in place. The best strategy is not obvious, again imposing an arms race to discover the best strategy in the presence of many agents attempting to do the same.

Technical details:

- **Attack range** is defined by l1 distance: "1 range" is a 3X3 grid centered on the attacker. Range 2 is a 5X5 grid, etc.
- **Incentive:** Animals hunt each other out of hunger or to protect/seize resources. Agents receive food/water resources equal to the damage they inflict on other agents.
- **Spawn Killing** Agents are immune during their first 15 game ticks alive. This prevents an exploit known as "spawn killing" whereby players are repeatedly attacked immediately upon entering the game. Human games often contain similar mechanism to prevent this strategy, as it results in uninteresting play.

## 7.4 API

We provide two APIs for this:

**Gym Wrapper** We provide a minimal extension of the Gym VecEnv API (Brockman et al., 2016) that adds support for variable numbers of agents per world and at any given time. This API distributes environment computation of observations and centralizes training and inference. While this standardization is convenient, MMOs differ significantly from arcade games, which are easier to standardize under a single wrapper.The Neural MMO setting requires support for a large, variable number of agents that run concurrently, with aggregation across many randomly generated environments. As such, the Gym framework incurs additional communications overhead in our setting that we bypass with a fast native API.

**Native** This interface is simpler to use and pins the environment and agents on it to the same CPU core. Full trajectories run locally on the same core as the environment. Interprocess communication is only required infrequently to synchronize gradients across all environments on a master core. We currently do the backwards pass on the same CPU cores for convenience, but plan on separating this into an asynchronous experience queue for GPU workers.

## 7.5 RENDERER

We also provide a front end visualizer packed with research tools for analyzing observe agent policies and relevant statistics. We leave detailed documentation of this to the source release, but it includes:

- 2D game renderer: shown in Figure 2
- 3D game renderer in beta
- Value map "ghosting" (See Figure 2)

- Exploration maps
- Interagent dependence
- Attack maps (See Experiments)

Preprocessor:

- Embed indicies corresponding to each tile into a 7D vector. Also concatenates with the number of occupying entities.
- Project visible attributes of nearby entities to 32D
- Flatten the tile embeddings
- Max pool over entity embeddings to handle variable number of observations
- Concatenate the two
- This produces an embedding, which is computed separately for each head.

Technical details

- For foraging experiments, the attack network is still present for convenience, but the chosen actions are ignored.
- It then concatenates the resultant vector with the (unlearned) features of the current entity. These include visible properties such as health, food, water, and position.
- In order to handle the variable number of visible entities, a linear layer is applied to each followed by 1D max pooling. Attention Bahdanau et al. (2014) may appear the more conventional approach, but recently OpenAI demonstrated that simpler and more efficient max pooling suffices.
- The pooled entity vector is then concatenated with the tile and self embeddings to produce a single vector and followed by a final linear layer as shown in Figure 10.

