# OpenReview forum: "Neural MMO: A massively multiplayer game environment for intelligent agents"
_ICLR.cc/2019/Conference_

### Official Review · AnonReviewer1 · 2018-11-02
**Revised Review (Score revised up to 7, from 5)**

**Rating:** 7
**Confidence:** 5

**Review:**

Revised Review:

The authors of this work has taken my concerns, and concerns of other reviewers, and revised their paper during the rebuttal period. They have increased the quality of the writing / clarity, restructured the presentation (i.e. put many details in the Appendix section), and committed to open-sourcing the platform post publication. For these reasons I believe this work is now at a state that should be published at ICLR, and I revised my score from 5 to 7. I hope other reviewers can reread the work and post their updated comments.

I'm excited about the work, because it incorporates good ideas from A-Life / evolution / open-endedness communities, to introduce new paradigms and new ways of thinking to the RL community. I look forward to using this environment in my own research going forward, regardless of whether this work gets accepted or not. Good luck!

Minor comment: On page 4, the section 5 Experiments, I think "Technical details" should be in bold font before the sentence "We run each experiment using 100 worlds." so it is distinguished from being part of that sentence.

Original Review:

The authors present a new game environment inspired by MMORPGs. The environment supports a "massive" number of agents, each have different neural net brains (*) and have some foraging and combat skills. They use distributed RL to train the policies (using REINFORCE) and over time can observe the dynamics of the population of these artificial life agents interact with each other, where the only reward is survival. There are many interesting insights, such as looking at how multi-agent cooperative (and deceptive) strategies emerge, and how some agents with different niche skills co-evolve with agents with other niche skills. They also plan to open source the platform and I have high hopes that this will be a fantastic research environment. While I'm very optimistic about this work and direction, there are issues with this particular paper, and I feel it is not ready for publication in its current form. While I have no doubt that the software project will be great, as a reviewer I'm evaluating this particular paper, and I want to highlight flaws about the paper and what can be done to fix it during the rebuttal/review period.

My recommendations to improve the article:

(1) Writing - I really enjoyed this work, but frankly, the writing is horrible. It took me days of effort to decipher every paragraph and understand all the terms and what is going on. The article reads like it is written by the person who programmed the game, and played MMORPGs almost every day of his childhood and adult life, so someone who is not reading the article thru the lens of the author might have an incredibly tough time digesting the content. For instance, there are sentences like "It adds melee, ranged, and magic-based combat"... "Melee, ranged, and magic combat have maximum Manhattan distance of effect of 1, 2, and 3 cells respectively. They do 10, 2, and 1 damage respectively"... "This prevents uninteresting 'spawn killing' and is a common technique in human games". These are only a small selection of samples. There are also terminology like "#ent and #pop" which I feel should be replaced by $N_{ent}$ and $N_{pop}$ for a paper. In contrast, older works related to population-based RL training like [2], or RL in games like [3] are examples of clear and understandable writing. I highly recommend you give the draft to someone outside of your team, who is sufficiently isolated from this project (or perhaps to a professional writer if your lab has one), to go over each paragraph, and make the writing more clear. This would benefit the work in the long term as people refer back to the paper when they run your code.

(2) Diagrams - While the diagrams look interesting, IMO they are poorly made. When I look at Figure 1, 4, and 9, it is really difficult to understand what is going on. I recommend redoing the diagrams, perhaps get some inspiration from distill.pub or OpenAI blog posts. There are things that are not clear, like what the inputs are into each agent, and how the training works. I recommend having some pseudocode snippets (like the Gym framework) to explain parts of the overall picture in more detail as figures.

Given a work of this magnitude, I'm personally okay that they went over 8 pages, as long as it is properly used for clarity.

Discussion:

Concepts from Artificial Life and Evolution has been introduced in this work. There is some confusion between what is "learning" and what has been "evolved" in your setup. Some readers coming from the evolution, or biology fields (who I bet will find your paper interesting to read and experiment with) might interpret "learning" to be weight changes during a life time, while "evolution" would be changes to the weight parameters from one generation to the next, but I think in policy-gradient RL, "learning" means weight changes after an agent dies and is reborn. Should consider clarifying in the introduction, the definition of learning, and whether it is inter-life or intra-life.

You cited some of Stanley's talks on open-endedness, but I wonder if you considered their work [1] where they proposed that having a minimum criteria condition might encourage diversity of solutions. For instance, perhaps in your environment, an agent doesn't have to be the very best, but only manage to survive, to move on to the next generation, which might cause very interesting multi-agent population dynamics. A parallel to modern life is that people (at least those in wealthy nations) live with such a good social safety net that people don't really have to be the best "agent" to reproduce and survive, and this might explain the large diverse cultures and ideas we end up with as human species, compared to other animals (where the current game is probably a suitable model of). An experiment to explore an experiment where only the very weakest agents die, but leaving agents with mediocre foraging and combat skills still live on (and pursue their own interests, whatever they may be) will be super interesting, and I encourage you to explore these ideas of open-endedness.

Bugs: In the appendix, the citation for OpenAI Five needs fixing.

Currently it pains me that I can only assign a score of 5 of this work (NOTE: this has been since revised upwards to 7 upon reading revision after rebuttal period), since I don't think the current writing is up to standards. In my opinion, it deserves a score of 7-8. If you work on points (1) and (2) and submit a revised draft with much better writing, visualization, figures to explain the work, I'll happily revise my score and improve it by 1-3 points depending on how much improvement is made.

[1] Brand and Stanley. "Minimal Criterion Coevolution: A New Approach to Open-Ended Search" (GECCO 2017) http://eplex.cs.ucf.edu/papers/brant_gecco17.pdf
[2] https://arxiv.org/abs/1703.03864
[3] https://arxiv.org/abs/1804.03720

(*) well, sort of, due to compute limits they are clustered to some extent to their species, so agents within a species have identical brains, unlike the real world.

---

> ### Author Response · Authors · 2018-11-24
> **Major revisions addressing all reviewer concerns**
>
> Thank you for your detailed and thorough commentary on our work. We appreciate the amount of effort you put into your review. You are absolutely correct: the initial writing needed a lot of work, and author proximity to the game genre behind the environment certainly played a part in the obfuscation. We have undergone a major rewrite and restructuring of the text and figures
>
> Much of the initial confusion among all reviewers surrounds minor implementation details that we scattered throughout the initial submission, such as melee/ranged/mage, attack ranges/damage values, and minor spawning mechanic details. These have been pruned, aggregated, and placed appropriately in a Framework appendix. This has also cut the paper to eight pages.
>
> The diagrams also do needed improvement. We have redone every figure in the main paper to at least include better captions, labeling, and color bars for the quantities being measured, and will continue to iterate on the visualizations in future drafts of the paper. It is amusing that you mention Gym usage snippets because our environment is callable through an almost identical API. We have included additional details in the framework appendix, but would provide full documentation with the open source release pending publication.
>
> Thank you for clarifying a potential source of confusion surrounding learning. Our experiments are consistent with your characterization of policy gradient RL, though our framework supports both definitions. We have clarified this in the writing.
>
> Your final suggestion is a phenomenal idea! Open ended task learning is precisely the long term objective of our work. We have tried to abstain from discussion of philosophy in the text of our paper because the base game environment does not yet provide a sufficiently general setting for varied and compelling problem solving. However, as our environment grows to better match the scale of human MMOs, we predict that agent behavior will diverge to fill a large space of reasonable but perhaps not entirely optimal strategies, as occurs among human players of MMOs.

---

### Official Review · AnonReviewer2 · 2018-11-02

**Rating:** 5
**Confidence:** 2

**Review:**

The paper presents a new evaluation platform based on massive multiplayer games, allowing for a huge number of neural agents in persistent environments.
The justification evolves from MMO as a source of complex behaviours arguing that these settings have some characteristics of life on earth, being a “competitive game of life”. However, there are many combinations with completely different insights and implications. The key characteristics for the setting in this paper seem to be:
1.	Cognitive evolution with learning, rather than physical or just genetic evolution (all bodies and architectures are equal)
2.	Changing environments (tasks), between parameter updates
3.	Survival-oriented rewards
And for some experiments some agents share policy parameters to simulate “species”.
From the introduction and the rest of the paper, it’s not clear whether the same platform can be used with agents that are not neural, or even agents that are hardcoded (for the sake of diversity or to analyse specific behaviours). This is an important issue, as other platforms allow for the definition of some baseline agents, including random agents, agents with simple policies, etc.
The background and related work section covers MMO and artificial life, but has some important omissions, especially those ideas in the recent literature that are closest to this proposal.
First, why can’t Yang et al., 2018 be extended with further tasks?
Second, conceptually, the whole setting is very similar to the Darwin-Wallace setting proposed in Orello et al. 2011:
@inproceedings{hernandez2011more,
  title={On more realistic environment distributions for defining, evaluating and developing intelligence},
  author={Hern{\'a}ndez-Orallo, Jos{\'e} and Dowe, David L and Espa{\~n}a-Cubillo, Sergio and Hern{\'a}ndez-Lloreda, M Victoria and Insa-Cabrera, Javier},
  booktitle={International Conference on Artificial General Intelligence},
  pages={82--91},
  year={2011},
  organization={Springer}
}

The three characteristics mentioned before are the key elements of this evaluation setting, which changes environments between generations. Also, the setting is presented in the context of evaluation and experimentation, as this manuscript.

Third, regarding multi-agent evaluation setting, Marlo over Minecraft (Malmo) is covering this niche as well.

https://marlo-ai.github.io/

Although it is episodic and the number of agents is limited, this should be compared too.

Nevertheless, the authors should make a more convincing argument about why we need *massively* multiplayer settings. Why is it the case that some behaviours and skills appear with thousands of agents but cannot appear with dozens of examples? In evolutionary game theory, for instance, some complex situations emerge from very few agents.

Finally, the use of agents that have to survive with “health, food and water” and its use as experimental setting can be found in Strannegård et al. 2018.
https://www.degruyter.com/downloadpdf/j/jagi.2018.9.issue-1/jagi-2018-0002/jagi-2018-0002.pdf
Figures are not very helpful. Especially the captions do not really explain what we see in the figures. For instance, Figure 2 doesn’t show much. Figure 3 left and middle show some weird dots and patterns, but they are not explained. Also, the one on the right tries to show “ghosting”, but colours and their meaning are not explained. Similarly, it is not clear what the agents see and process. I assume it is a local grid as the one seen in figure 4. But this is quite an aerial view, and other grid options might do the job as well.
Similarly, some actions are mentioned (it seems that N, S, E, W and “Pass”? plus some attack options, but they are not described). In the end, I understand many choices have to be made for any evaluating setting, but many choices are very arbitrary (end of section 3 and especially experiments) and there is a lot of tuning, so it’s unclear whether some of the observations happen just in a particular combination of choices, but are more general. The authors end up with many inconclusive observations and doubts (“perhaps”) about small changes, at the end of section 5.
Other things such as the “spawn cap” and the “server merge” are poorly explained, with clear definitions and proper justification of their role. Similarly, I’m not sure about how reproduction takes place or not, and if so, whether weights are inherited or reinitialised. Something related is said about species.
I found the statement about multiagent competition being a curriculum magnifier, not a curriculum itself, very interesting, but is this really shown in the paper or elsewhere?
In general, I miss many details and justifications for the whole architecture and mechanism of this neural MMO.
Pros:
-	Designed to be scalable
-	Goes in the right direction of benchmarks that can capture generally variable (social) behaviour.
Cons:
-	Poor comparison with existing platforms and similar ideas.
-	Too many arbitrary decisions for the setting and the experiments to make it work or show complex behaviours
-	The paper needs extensive rewriting, clarifying many details, with the figures really helping for the understanding.
Typos and minor things:
-	“Susan Zhang 2018” is named a couple of times, but the reference is missing. Also, it is quite unusual to use the given name for this researcher while this is not done for any other of the references.
-	“as show in Figure 2” -> shown
-	“impassible” -> “impassable”

****************************
I've read the new comments from the authors and the new version of the paper. I think that the paper has improved significantly in terms of presentations, coverage of related work. I still see that the contribution is somewhat limited, but I'm updating the score to better account with this new version of the paper.

---

> ### Author Response · Authors · 2018-11-24
> **Major revisions addressing all reviewer concerns**
>
> Thank you for your commentary—your recommendations surrounding comparison to related work has allowed us to better define the space that our work fills. The original writing was admittedly confusing; we have undergone a major restructuring and rewording of the paper in order to address your criticisms and those of the other reviewers:
>
> - The framework assumes only that agents receive information about their local environment and output decisions. Hardcoded and algorithmic agents are supported in the framework. While our work is closer to a platform than a fixed task, random baseline agents live for 10-20 timesteps on average.
> - The main difference from the four works you mention is that our environment is modeled after the real game genre of MMOs. This combines much of the interpretability of earlier game environments, such as Atari, with forms of large scale multiagent interaction unavailable in prior game environments. We have added additional citations.
> - We do not argue that massively multiplayer interaction is needed in all settings; the purpose of this work is precisely to investigate the types of behaviors that do emerge in such a setting.
> - All figures have been recaptioned and augmented with scales for the quantities measured. These were clearly confusing given that it was not obvious that the weird dot patterns you mention are actually the agents.
> - Input and action spaces of the environment are now described in full detail.
> - “Tuning” in our setting is simply game development.  Our goal was to produce an initial set of design choices that support interesting multiagent interactions and avoid trivial behavior.
> - All classic game terms such as “spawn cap” and “server merge” are toned down and clearly defined when used.
> - Multiagent competition as a curriculum magnifier is a driving paradigm behind our platform; the argument for this is now detailed in Discussion.
> - A better description of MMOs in the context of this work has been added.

---

### Official Review · AnonReviewer4 · 2018-11-13
**Interesting ideas and results but lacking clarity and focus.**

**Rating:** 6
**Confidence:** 4

**Review:**

This paper proposes a multi agent life simulators as an environment for RL. The environment is procedurally generated, with possibly many different game dynamics including foraging, and combat. They train deep RL agents in this environment and show various emergent behaviors such as exploration, and niche development. Additionally, they propose a tournament competition scheme to evaluate different populations of agents against each other.

This paper has a number of interesting findings, but overall lacks polish and coherence. The writing is verbose and informal in many places. There are many details not included -- for example specifics on combat targeting, how RL agents are trained, and information how to parse figures (what do colors mean?).

Pro
+ Interesting idea, and demonstration of a system. From the intro, I believe an environment such as this is will be fruitful to study.
+ Results seem preliminary but are interesting. In particular, the finding that agents generalize and thus perform better on tournament selection when trained in larger population is intriguing as well as the exploration results with population count!
+ Reproducibility: authors claim they will release environment simulator code.

Con
- Paper can be considerably tightened. It is currently quite long (9.5 pages vs the suggested 8 page). There are also a lot of details included that don't seem core to the message. For example -- the multiple types of API / IPC communication, much of the RPGs section.
- Some areas of writing could be improved, either too casual, or sloppy. For example -- various names are not capitalized in bibliography.
- Examples of imprecise / casual writing: "good performance without discounting, but training was less stable." What does "less stable" actually mean? "postprocess trajectories using a discount factor" this is part of the REINFORCE algorithm -- postprocessing, to me, implies modifying the observations. The term "numerical collapse" is not a term I am aware of.
- It is unclear what is shown in many of the figures. What are the colors in figures 8,9,10 for example?
- Lots of details and ongoing work put in which distracts from a clear message. For example, why was "entity targeting" included? It doesn't appear to be described and the results shown in figure 10 are confusing. I would consider stepping back, and figuring out what one thing you want to show the reader, then drop all detail not around that point.
- Lacking a conclusion of somesort. Ideally there would be something to pull the whole paper together.
- use of terminology -- unclear why neural mmo is name of this environment. This is not a MMO, nor does the environment have anything "neural" related -- one can train reinforcement learning agents without neural network function approximators on it for example. I would consider renaming.

In its current form, I do not recommend accepting this paper but I do encourage the authors to continue working on it to both tighten the writing and presentation as well as continue to show interesting results via RL experiments.



EDIT: See bellow, raised score from 4 --> 6.

---

> ### Author Response · Authors · 2018-11-24
> **Major revisions addressing all reviewer concerns**
>
> Thank you for your detailed list of revision suggestions. Agreed—the original writing was a mess! Many of the details you were looking for were present, but scattered throughout the paper in a less than comprehensible manner. Here is a non exhaustive summary of improvements as per your comments:
>
> - Writing tightened to 8 pages. Many details important to documentation but not core to the message have been moved to a Framework appendix.
> - Clarified exact training details
> - Simplified and added relevant scales to all figures
> - Moved all details of ongoing work to discussion and additional insights in order to isolate them from matured experiments.

---

> ### Author Response · Authors · 2018-12-06
> **Review Updates**
>
> Please let us know if we can provide any additional information to aid in reevaluating our work -- we have revised the paper to address all of your major concerns. We believe the paper is much improved, and both other reviewers have updated their evaluation.

---

> > ### Comment · AnonReviewer4 · 2018-12-10
> > **Review of revised paper.**
> >
> > First, sorry for the delay on this. I missed the original open review email and didn’t realize there was an update. Thank you for the ping. I have done a complete reread and have a second review bellow.
> >
> > Pro: (from last version)
> > You are correct that the paper greatly improved. Many confusing points have been resolved. The underlying story of building large scale environments and the experiment results remain interesting and are easier to follow.
> >
> > Cons:
> > The combat and targeting system are still confusing. In fact, I don’t think they are particularly core to the story and could even be expanded and moved to the appendix. Targeting is not really define. Consider adding a diagram showing what this is. I have a guess from playing RTS style games, but not 100% sure. You state learned targeting does not work yet provide figures about it. I also wouldn’t recommend including these results as they detract from the main story.
> >
> > Figure 8  it is unclear to me what the difference between the two maps on the right side are.
> >
> > Also, because I think it would be interesting and informative, it would be great to include a video of what “life” looks like at multiple different resolutions.
> >
> > Try not to present new results in the discussion.  Right now the discussion section reads more like an experimental section. I would merge all results, and write a discussion that is more big picture about this type environment, limitations, and future work. Right now, 1-5 are great, clear, tell a nice story and are self contained. The message becomes a foggier with the discussion section — doesn’t fit the story and seems like a random other grab bag of experiments.
> >
> > In light of improvements, and given that this work was interesting to read, and exploring an undeveloped area of research, I am going to change my review by 2 pt: 4=>6. I do hope the authors continue to do work on this and improve the messaging around the paper as this will ensure larger impact!
> >  Typos:
> > “Technical details We run each experiment using 100”
> > Delete technical details? Is it meant to be a header?
> >
> > “We train for a fixed number of trajectories per population 9.”
> > What is the 9 for? Is this a typo?

---

### Meta-Review · Area_Chair1 · 2018-12-20

**Confidence:** 4
**Recommendation:** Reject

**Metareview:**

The reviewers raise a number of concerns including limited methodological novelty, limited experimental evaluation (comparisons), and poor readability. Although the authors did address some of the concerns, the paper as is needs a lot of polishing and rewriting. Hence, I cannot recommend this work for presentation at ICLR.